# Optimizing hypertension prediction using ensemble learning approaches

**Isteaq Kabir Sifat, Md. Kaderi Kibria**[ID]*

Department of Statistics, Hajee Mohammad Danesh Science and Technology University, Dinajpur, Bangladesh

* kibria.stt@tch.hstu.ac.bd

**Data Availability Statement:** https://figshare.com/s/a709a390ecd276046607?file=41735691.

**Funding:** The author(s) received no specific funding for this work.

## Abstract

Hypertension (HTN) prediction is critical for effective preventive healthcare strategies. This study investigates how well ensemble learning techniques work to increase the accuracy of HTN prediction models. Utilizing a dataset of 612 participants from Ethiopia, which includes 27 features potentially associated with HTN risk, we aimed to enhance predictive performance over traditional single-model methods. A multi-faceted feature selection approach was employed, incorporating Boruta, Lasso Regression, Forward and Backward Selection, and Random Forest feature importance, and found 13 common features that were considered for prediction. Five machine learning (ML) models such as logistic regression (LR), artificial neural network (ANN), random forest (RF), extreme gradient boosting (XGB), light gradient boosting machine (LGBM), and a stacking ensemble model were trained using selected features to predict HTN. The models' performance on the testing set was evaluated using accuracy, precision, recall, F1-score, and area under the curve (AUC). Additionally, SHapley Additive exPlanations (SHAP) was utilized to examine the impact of individual features on the models' predictions and identify the most important risk factors for HTN. The stacking ensemble model emerged as the most effective approach for predicting HTN risk, achieving an accuracy of 96.32%, precision of 95.48%, recall of 97.51%, F1-score of 96.48%, and an AUC of 0.971. SHAP analysis of the stacking model identified weight, drinking habits, history of hypertension, salt intake, age, diabetes, BMI, and fat intake as the most significant and interpretable risk factors for HTN. Our results demonstrate significant advancements in predictive accuracy and robustness, highlighting the potential of ensemble learning as a pivotal tool in healthcare analytics. This research contributes to ongoing efforts to optimize HTN prediction models, ultimately supporting early intervention and personalized healthcare management.

## Introduction

Hypertension (HTN), characterized by elevated blood pressure beyond normal ranges, is a major public health issue affecting adult worldwide [1, 2]. It significantly increases the risk of

**Competing interests:** The authors have declared that no competing interests exist.

**Abbreviations:** HTN, hypertension; HHTN, history of hypertension; ML, machine learning; LR, logistic regression; ANN, artificial neural network; RF, random forest; XGB, extreme gradient boosting; LGBM, light gradient boosting machine; NB, naïve base; AUC, area under the curve; BMI, body mass index; SHAP, SHalpey Aditive exPlanaions; WHO, World Health Organization; DT, decision Trees; NB, naïve bayes; BFS, Boruta-based feature selection; LASSO, Least Absolute Shrinkage and Selection Operator; ADASYN, Adaptive synthetic; CRT, classification and regression task; OR, odds ratio; GOSS, gradient-based one-side sampling; EFB, exclusive feature bundling.

cardiovascular disease, coronary heart disease, stroke, kidney damage, and other serious complications if left untreated [3, 4]. HTN is the leading causes of premature death globally and affects more than one in four men and one in five women [4]. Due to its high prevalence and association with chronic kidney disease, HTN poses a major global health challenge [5–7]. As a primary risk factor for cardiovascular diseases, it contributes to rising healthcare costs and loss of productivity [8]. According to the World Health Organization (WHO), HTN is the third greatest cause of death worldwide, responsible for one in every eight fatalities [9]. Currently, an estimated 1.3 billion people globally, including 116 million Americans, live with HTN [10]. Hypertensive individuals face a 2–4 times greater risk of developing heart disease, peripheral vascular disease, and stroke [11, 12], which exacerbate the economic burden of out-of-pocket expenses and contribute to the leading causes of morbidity, mortality, and disability [13–15]. By 2025, approximately 1.56 billion adults aged 30 to 79 are projected to have HTN, with nearly two-thirds of them residing in low and middle income countries [16]. Given the high prevalence, controlling and predicting HTN at earlier stage is crucial. Early identification of interpretable risk factors in HTN patients is crucial for enabling timely prevention and intervention. Thus, detecting, diagnosis and understanding the risk factors associated with HTN are critical for effective management and treatment.

Modeling to predict the risk of acquiring HTN can aid in identifying significant risk factors contributing to HTN, offering reliable estimates of future HTN risk [17], and identifying individuals at high risk who may benefit from medical care and adopting healthy behaviors to prevent HTN [18–20]. Numerous prediction models have been created over time to forecast the risk of HTN in the general population. Models were created using either a contemporary machine learning technique or a conventional regression-based approach [21]. Primarily, prior research has examined conventional linear models, like logistic regression (LR) and the Cox proportional hazard model, in order to determine the risk factors significantly linked with HTN [22–24]. Also, several studies applied some machine learning techniques and measure their HTN prediction accuracy. A ML-base prediction study has explored a plethora of algorithms, with RF, K-Nearest Neighbors (KNN), DT, and Naive Bayes (NB) models showcasing promising results, with RF boasting an impressive accuracy of 80.12% [25]. In a subsequent study, RF, CatBoost, MLP Neural Network, and LR were used, with RF achieving an accuracy of 92% [26]. Medical data-based study, SVM, C4.5, RF and XGBoost methods were applied and 94.36% accuracy was achieved by using the XGBoost method [27]. However, a study utilizing RF, LR, ANN, and XGBoost models on Ethiopian data yielded a comparatively modest accuracy of 88.81% for XGBoost [28], which is lower than that reported in previous studies. Improving prediction performance for HTN data is therefore necessary. To enhance the performance of these classifiers, several strategies can be employed. Recent research has highlighted the efficacy of the Light Gradient Boosting Machine (LGBM) learning algorithm, which utilizes tree-based learning techniques and has outperformed existing machine learning algorithms [29]. Consequently, this study aims to apply this updated algorithm and their stacking model approach to enhance prediction accuracy using Ethiopia data, to advance the understanding and predictive capabilities in combating HTN. Although stacking models are well-established in machine learning, our study applies these methods in the specific context of HTN risk prediction, an area where their potential remains underexplored. The key contribution of our work lies in demonstrating how integrating multiple models through an ensemble approach significantly boosts predictive accuracy compared to single-model methods. Although the theoretical foundation of ensemble learning may not be novel, the improvement in predictive accuracy and its potential for real-world clinical application represents a meaningful contribution to the field.

## Materials and methods

### Data collection and data processing

This study utilized secondary data on HTN collected from Paulose et al., (2022), where they investigated the prevalence and associated factors of HTN among the population of Hawassa City in Ethiopia [30]. The original study was a cross-sectional study carried out in the community by the Hawassa city administration. Participants had to be at least 30 years old and have resided in the city for a minimum of six months to be included. A total of 612 samples were collected using a multi-stage sampling technique. It was selected to represent diverse demographic and clinical characteristics associated with HTN risk in the Ethiopian population. Data collection involved the use of a structured questionnaire that captured demographic and socio-economic variables, as well as information on blood pressure history, co-morbidities, behavioral factors, and physical measurements. Blood pressure measurements were taken for all 612 participants to confirm the presence or absence of HTN. HTN diagnosis followed the WHO standard (systolic pressure at least 140/90 mmHg and/or diastolic pressure at least 90 mmHg) and was conducted by trained nurses. Individual risk factors for hypertension were identified based on different levels of explanatory variables, and the quantitative variables were classified according to prior sessions (see **S1 Table**) [28, 31–33]. The collected dataset didn't contain any missing values or outliers, as these issues were addressed during the original data collection process. Therefore, no additional steps for data cleaning, outlier detection, or imputation were required. The completeness and quality of the dataset ensured that the data was ready for analysis, enhancing the reliability of our results.

### Feature selection

Feature selection techniques are essential in ML, as they allow for the extraction of the most relevant attributes for classification [34]. This not only improves the performance of the model during training but also facilitates easier interpretation of the model's outcomes [35]. To determine the most critical subset of features, we applied four feature selection algorithms: Boruta-based feature selection (BFS) method, Least Absolute Shrinkage and Selection Operator (LASSO) regression, Forward and Backward Selection (FBS) and random forest (RF). Boruta is a wrapper-based feature selection method that utilizes the RF classifier algorithm, renowned for its unbiased and robust performance [36]. The LASSO algorithm introduces *L1* regularization to the regression model, penalizing the number of features to prevent overfitting [37]. Random Forest entails constructing multiple decision trees based on bootstrap samples [38]. Stepwise methods iteratively enhance the selected variables by including or excluding one variable at a time; examples include forward selection and backward selection algorithms [39]. After applying each feature selection approach, we intersected the results of all four methods to identify the most significant risk factors associated with hypertension (HTN).

$$Common\ features = \bigcap_{i=1}^{r} Identified\ features\ form\ HTN\ dataset_i$$

Where, $r$ is the number of utilizing feature selection methods (here, $r = 4$).

### Machine learning algorithms

The study employed six machine learning algorithms to achieve its objectives, each chosen for their specific strengths in handling prediction tasks (see S1 Appendix for details). Logistic Regression (LR), a widely used supervised algorithm, was selected for its ability to predict the probability of binary outcomes, which aligns with the study's goal of distinguishing between

success and non-success outcomes [40, 41]. RF were chosen for their ability to improve decision trees by using bagging to build multiple trees, addressing overfitting through averaging predictions [40, 42–44]. ANN simulates the human brain's reasoning and pattern recognition, offering robust methods for identifying complex relationships [45, 46]. XGBoost, a gradient-boosting algorithm, was included for its efficiency in predicting residuals from previous models, contributing to the overall predictive power [47, 48]. Similarly, LGBM, known for its speed and accuracy, was selected for its histogram-based approach and leaf-wise strategy, making it effective in large-scale data processing [49–51]. The stacking technique was employed as a meta-learning approach, where predictions from these diverse base models were used to train a new meta-learner, enhancing the overall performance [52]. The stacking classifier consisted of LR, ANN, RF, LGBM and XGBoost as base classifiers (level 0), with LR as the meta-learner (level 1). LR was chosen as the meta-learner due to its simplicity and interpretability, providing clear explanations of how base models contribute to final prediction [53]. This approach also mitigates overfitting in complex ensembles. While LR is effective in avoiding overfitting, its assumption of linearity may limit flexibility in datasets with non-linear relationships. In such cases, more complex meta-models like Gradient Boosting or RF could capture these intricate patterns more effectively, though they may introduce a higher risk of overfitting if the base models are already complex. The combination of these algorithms ensured that each model contributed meaningfully, while weaknesses in one algorithm were compensated for by the strengths of others, improving the overall accuracy and robustness of the predictions.

## Data partition and balancing

The entire dataset was randomly split into two sets: 70% was used for training (HTN: 21.2%, non-HTN: 78.8%) and 30% was used for testing (HTN: 21.3%, non-HTN: 78.7%) and employing a technique for stratified sampling. Due to the class imbalance in the data, the majority class of the target variable may lead to biased results in classification tasks. To address this issue; several data balancing strategies can be employed. In our study, we used the Adaptive synthetic (ADASYN) balancing approach alongside under-sampling to balance the training set. Although our sample size may appear limited for machine learning applications, the ADASYN technique helped mitigate class imbalance and reduce potential overfitting. A larger dataset would improve the robustness and generalizability of the findings, and plans are in place to expand the dataset in future research or collaborate with larger cohorts to enhance the credibility and external applicability of the results.

## Cross-validation and tuning hyperparameters

There are additional parameters, referred to as hyperparameters, in the ML algorithms discussed above. To increase the model's performance, the user can explicitly define hyperparameters before the learning process. The hyperparameter values in the training set were adjusted by the grid search technique using a repeating 10-fold ($K$10) cross-validation procedure. To carry out the $K$10 technique, a training subset and a verification set are separated from the training dataset in 7:3 ratios.

## Kernel SHAP-based interpretability method

Kernel SHAP is a method that computes SHapley values using a specialized weighted linear regression function to estimate each feature's contribution [54]. SHapley values account for the various magnitudes and signs with which risk factors influence the model's result or prediction. As a result, SHapley values represent estimations of the contribution's feature important amount and direction (sign). In the model, risk factors with a positive SHAP value help

predict patients with HTN, whereas those with a negative SHAP value assist in forecasting patients under control [28]. Specifically, the significance of every risk factor, let's say the $k^{th}$ risk factor, is determined by the SHapley value, which is determined by the following formula

$$\emptyset_k(v) = \frac{1}{M!} \sum_{S \subseteq M \setminus \{k\}} |S|!(M - |S| - 1)![v(S \cup \{k\}) - v(S)]$$

where, $S$ stands for the subset of risk factors that excludes the risk factor for which the value is being calculated; $\emptyset_k(v)$; $S \cup \{k\}$ is the subset of risk factors that includes the $k^{th}$ risk factor in S; v(S) is the result of the ML-based model that explains using the risk factors of S; $S \subseteq M \setminus \{k\}$ and represents every set of $S$ that is a subset of every risk factor set in all of $M$, which excludes the $k^{th}$ risk factor.

## Performance evaluation criteria

We assessed each method's prediction performance in terms of precision, sensitivity, and specificity. The following is a description of the equations:

$$Accuracy = \frac{TP + TN}{TP + FP + TN + FN}$$

$$Sensitivity = \frac{TP}{TP + FN}$$

$$Specificity = \frac{TN}{TN + FP}$$

where true positives, true negatives, false positives, and false negatives are denoted, respectively, by TP, TN, FP, and FN. Additionally, metrics such as the area under the curve (AUC) and the receiver operating characteristic (ROC) curve are also evaluated. The calculation formula of AUC is as follows:

$$AUC = \int_{x=0}^{1} TPR(FPR^{-1}(x))dx$$

## Ethical approval

To address the ethical considerations in this study, it is important to note that the data were sourced from a prior research project that received ethical approval from the Research and Ethics Committee of the University of South Africa (UNISA) (Reference number: REC-012714-039). The secondary analysis was conducted in compliance with ethical guidelines regarding the use of health-related data. As the original dataset was anonymized, no additional ethical approval was required for this analysis. We acknowledge the sensitivity of health-related information and have implemented measures to ensure participant confidentiality and data security throughout the study. Fig 1 illustrates the complete workflow of this study.

## Results

### Baseline characteristics

The study included 612 participants, of whom 130 (21.2%) had HTN and 482 (78.8%) did not. Among the participants, 53.4% were male, and over half resided in urban areas. The mean age of the cohort was 47.56 ± 13.40 years, with an average height and weight of 165.20 ± 8.87 cm and 66.589 ± 8.769 kg, respectively. HTN prevalence was notably higher among obese individuals compared to those with normal weight (50% vs. 13.4%). Additionally, individuals with

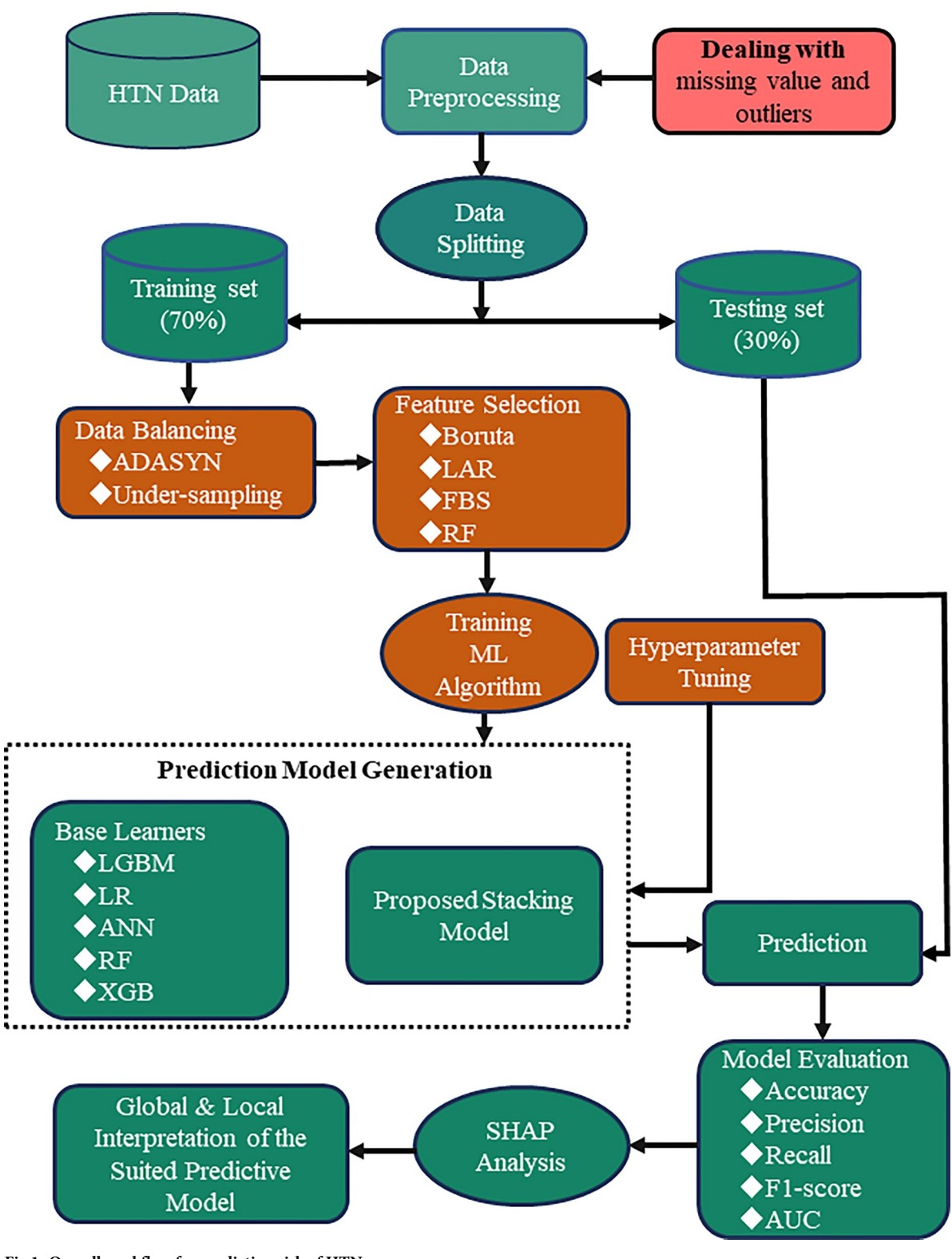

**Fig 1. Overall workflow for prediction risk of HTN.**

**Table 1. Names, descriptions, and categorizations of the 13 common features selected using four feature selection methods.**

| S/N | Name | Description | Categorization |
|---|---|---|---|
| 1. | Age | Age of the respondents | Continuous variable (yearly) |
| 2. | Education | Educational level | Can't read and write, Read and write only, Primary, Secondary, Diploma and above |
| 3. | Walking | Walking at least 10 minutes | Yes, No |
| 4. | Weight | Weight of the respondents | Continuous variable (kg) |
| 5. | BMI | Body mass status | Underweight, Normal, Overweight, Obese |
| 6. | Smoking | Smoking status | Yes, No |
| 7. | Drinking | Drinking alcohol | Yes, No |
| 8. | Vegetable | Eat fruit at least per week | Yes, No |
| 9. | Fat | Eating animal fat | Yes, No |
| 10. | Salt | Eating habit salt | Yes, No |
| 11. | Transport | Mode of transport | On foot/pedal bicycle, Engine |
| 12. | HD | History of diabetes | Yes, No |
| 13. | HHTN | History of hypertension | Yes, No |

diabetes (47.5% vs. 30.0%) and smokers (50.4% vs. 23.8%) exhibited higher prevalence rates of HTN. Having a family history of diabetes (41.8% vs. 11.2%) was linked to higher rates of HTN prevalence. Significant associations (P-value < 0.005) were observed between HTN and various factors including age, sex, residence, occupation, income, physical activity, diabetes, weight, height, BMI, smoking, alcohol consumption, dietary habits, transportation mode, and socioeconomic status (see **S1 Table**).

## Risk factors identification of HTN using different method

Feature selection involves identifying and removing inappropriate, irrelevant, or unnecessary features from a dataset to improve model accuracy. This study utilized four feature selection algorithms: BFS method, LASSO regression, FBS and RF, to identify important features that were common among these methods (see **Table 1**). The Boruta algorithm identified 18 features, while the forward-backward selection method identified 13, RF identified 19, and Lasso Regression identified 18. Each method revealed specific and important features associated with HTN. Thirteen features were found to be common across all four feature selection methods and were considered important risk factors for HTN (see **Table 1 and S2 Table**). These identified features are associated with hypertension, though their contribution to its development cannot be established based on the current data. The identified 13 risk factors were considered in the machine learning-based model for predicting HTN status.

## Performance comparison of ML-based models

The performance of various machine learning models using imbalanced data, under-sampling and ADASYN for predicting hypertension is shown in **Table 2** and **S3 Table**. The ADASYN technique significantly contributed to the improved performance metrics by addressing the class imbalance between hypertensive and non-hypertensive participants. By generating synthetic samples for the minority class (HTN), ADASYN enabled the model to better learn the patterns associated with hypertension, thereby reducing bias towards the majority class and enhancing predictive accuracy. The models' performance was evaluated using various metrics, including accuracy, precision, recall, F1-score, and AUC. The results indicated that the prediction performance of the ML models was comparatively low for imbalanced data (see **S3 Table**). However, for under-sampling and ADASYN data, prediction performance was comparatively better (see **Table 2**). The most accurate prediction was made by the LGBM model

**Table 2. Performance of the five ML models and their stacking models with two-class balancing.**

| Balancing Method | Models | Accuracy | Precision | Recall | F1-Score | AUC |
|---|---|---|---|---|---|---|
| Under Sampling | LGBM | 87.43 | 88.14 | 92.75 | 78.6 | 0.878 |
| | LR | 83.82 | 86.47 | 89.65 | 82.53 | 0.844 |
| | ANN | 86.34 | 81.54 | 93.06 | 84.08 | 0.879 |
| | RF | 82.51 | 86.84 | 94.48 | 53.01 | 0.777 |
| | XGB | 84.69 | 88.30 | 84.41 | 69.56 | 0.785 |
| ADASYN | LGBM | 92.61 | 91.24 | 94.12 | 92.46 | 0.943 |
| | LR | 86.53 | 83.01 | 90.07 | 86.39 | 0.912 |
| | ANN | 88.55 | 84.97 | 92.36 | 88.51 | 0.942 |
| | RF | 88.55 | 83.89 | 93.24 | 88.34 | 0.937 |
| | XGB | 90.61 | 92.05 | 90.25 | 91.14 | 0.933 |
| | Proposed Stacking | 96.32 | 95.48 | 97.51 | 96.48 | 0.971 |

using the ADASYN balancing technique with an accuracy of 92.61%, precision of 91.24%, recall of 94.12%, F1-score of 92.46% and AUC of 0.943 compared to other models. Additionally, to increase the prediction accuracy of HTN, we used stacking ensemble learning models. The results indicated that the best predictive discrimination ability was attained by our suggested stacking model with an accuracy of 96.32%, precision of 95.48%, recall of 97.51%, F1-score of 96.48%, and AUC of 0.971 compare to other five ML-models (see **Table 2**).

The ROC curves of the five predictive models and the proposed stacking model with ADASYN are shown in Fig 2. The ROC values also indicated that our proposed stacking model is significantly better than the LGBM, LR, ANN, RF and XGB models. Therefore, based on the prediction performance results, it is demonstrated that our proposed stacking model with ADASYN performed better.

### Interpretable hypertension risk factors

Using SHAP values, an in-depth analysis was conducted to identify interpretable predictive risk factors for HTN within the proposed stacking model. The SHAP summary plot provides a detailed view of how the input risk factors influence the predictions. The significance, impact, initial value, and connection of the risk factors to the high risk of HTN are illustrated by the swarm plot in Fig 3. The x-axis displays the direction and magnitude of the influence of each risk factor, with positive values suggesting an increased risk of HTN and negative values indicating a decreased risk. The color gradient represents the value of each risk factor, with red denoting high values and blue indicating low values. The analysis revealed that several factors are significantly associated with HTN risk such as weight, alcohol consumption, a history of HTN, salt intake, age, diabetes, BMI, and fat intake. High weight, excessive alcohol consumption, and a prior diagnosis of hypertension correlate strongly with increased HTN risk, while high salt intake and advancing age further elevate this risk. Additionally, a history of diabetes and elevated BMI contribute to susceptibility, consistent with established clinical knowledge. These findings underscore the importance of these risk factors in predicting HTN, suggesting targeted intervention strategies that could focus on weight management, dietary modifications, and lifestyle changes to mitigate the risk of developing HTN.

### Discussion

Hypertension prediction is vital for effective preventive healthcare strategies [55]. Assessing current HTN risk is essential for identifying underlying risk factors that may not yet be

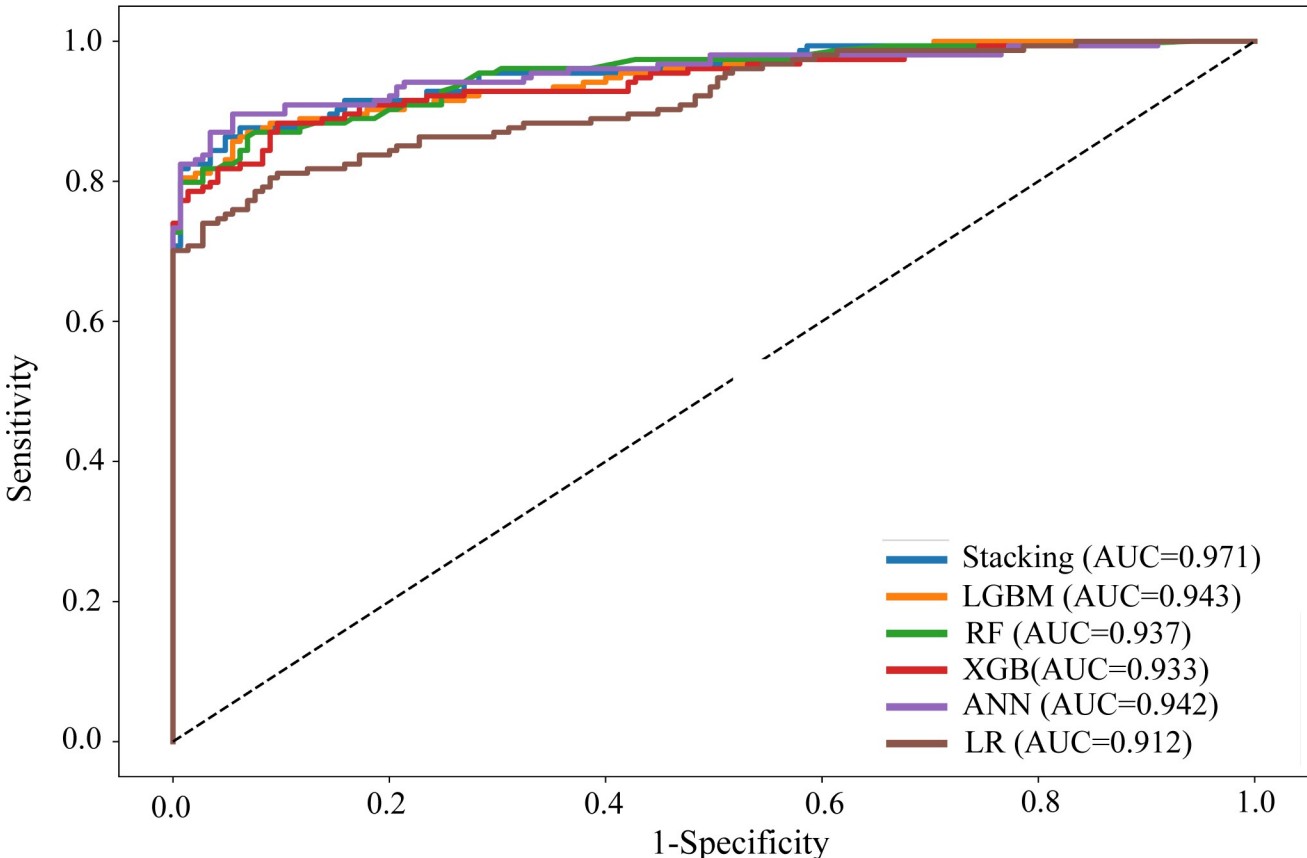

**Fig 2. ROC curves of the six predictive models.**

reflected in elevated blood pressure. While predicting future risk aids in early detection and intervention, elevating current risk helps healthcare providers identify individuals at risk of developing HTN. This proactive approach enables timely lifestyle modifications and interventions to prevent its progression. Integrating both current and future risk assessments enhances HTN management strategies, addressing immediate concerns while mitigating long-term risk. This study aimed to improve the prediction of HTN using ensemble learning approaches. By integrating multiple ML algorithms, we sought to improve the accuracy of predictive models compared to traditional single-model methods. Our research utilized a comprehensive dataset from Ethiopia, consisting of 612 participants with 27 features potentially associated with HTN risk. The dataset underwent various balancing techniques, including under-sampling and ADASYN, to address class imbalance issues. We employed a comprehensive feature selection approach that integrated BFS, LASSO regression, FBS, and RF feature importance to explore the most relevant predictors.

Using 13 risk factors identified through this technique, we trained five ML algorithms (ANN, LR, RF, XGB, and LGBM) as well as a stacking model to predict HTN. The performance of these models was evaluated on the testing set using metrics such as AUC value, accuracy, precision, recall, and F1-score. The accuracy of the five ML-based models with ADASYN are as follows: 92.61% for LGBM, 86.53% for LR, 88.55% for both ANN and RF, and 90.61% for XGB. Our proposed stacking model achieved an accuracy of 96.32%. According to these performance metrics, we recommend the stacking model as the optimal classifier for predicting HTN. A previous study on the same dataset showed prediction accuracies of 86.43% for

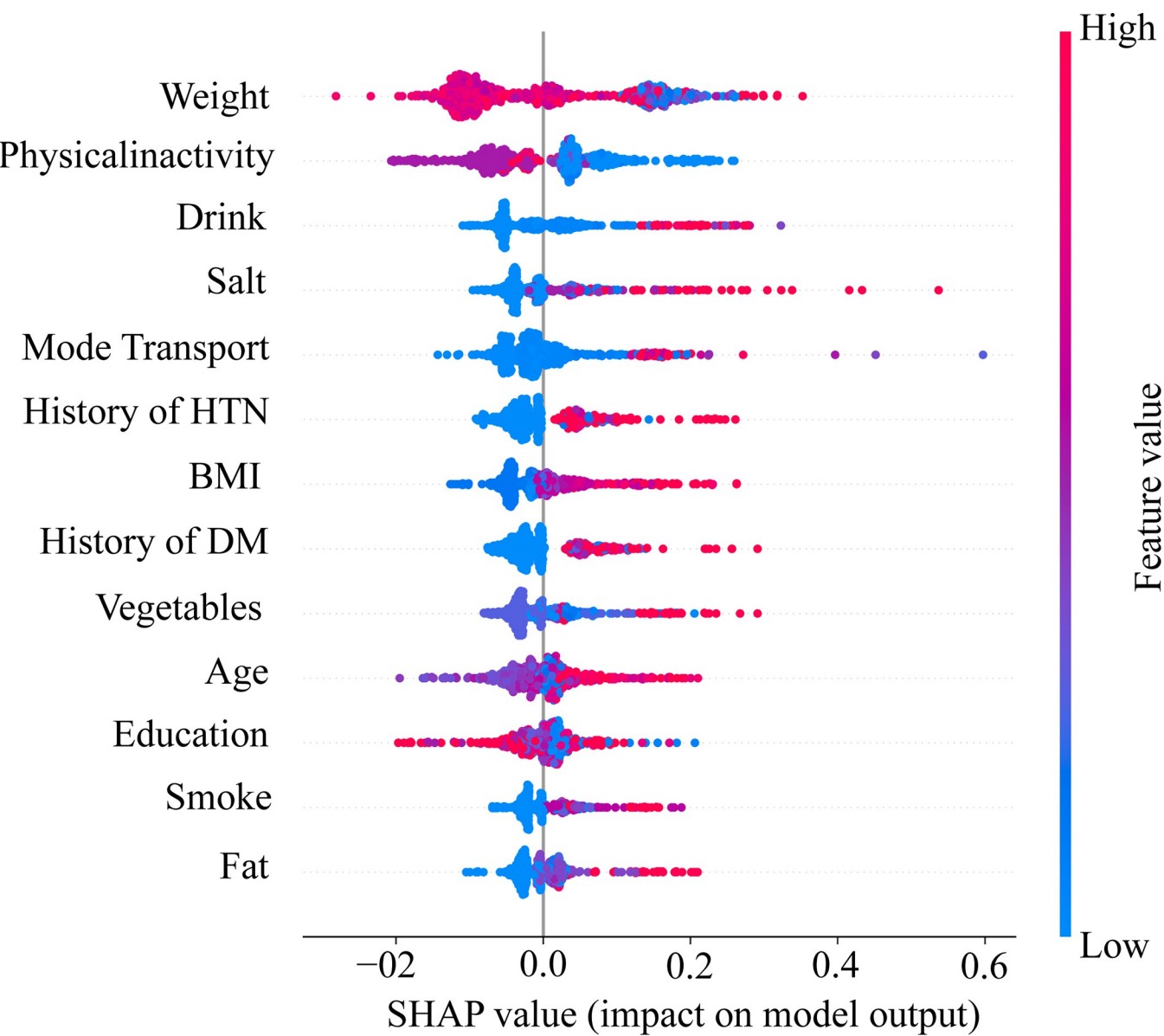

**Fig 3. SHAP values: Impact of risk factors on prediction outcome.**

LR, 85.25% for ANN, 87.88% for RF, and 88.81% for XGB [28]. The XGB ML-model achieved highest accuracy (88.81%) in this study which is almost similar with our result. Another study showed that LGBM is more robust than other ML models [29], achieving the highest accuracy (92.61%) for our HTN data. To further increase prediction accuracy, we used the stacking model, which achieved an accuracy of 96.32%, outperforming the existing single-model prediction accuracies [28]. Additionally, a study on HTN risk detection selected features using FI-FA and MC-FA models, with MC-FA (10 Factor) and algorithms like RF, KNN, DT, and NB showing promising results, particularly RF with an AUC of 85.96% and accuracy of 80.12% [25]. In subsequent study, the LR approach was used to identify significant HTN risk factors, alongside RF, CatBoost, LR and MLP Neural Network models. Among these, RF achieved an accuracy of 82% and an AUC of 0.92. [26]. Another study found that LR

Table 3. Comparison of prediction performance of exiting models for hypertension risk prediction.

| Reference | Country | Study type | Data size | No. of riskfactors | Best model | AUC | SHAP |
|---|---|---|---|---|---|---|---|
| Islam et al., 2023 [28] | Ethiopia | CS | 612 | 27 | XGB | 0.89 | yes |
| Zhao et al., 2021 [26] | China | CS | 29,700 | 11 | RF | 0.92 | No |
| Kurniawan et al., 2023 [56] | Indonesia | CS | 30,320 | 11 | LR | 0.829 | No |
| Wu Y et al., 2024 [57] | China | CS | 476 | 9 | RF | 0.95 | No |
| Chai et al., 2022 [58] | Malaysia | CS | 2461 | 11 | LGBM | 0.686 | No |
| Islam et al., 2021 [59] | Bangladesh | CS | 6965 | 13 | GB | 0.669 | No |
| AlKaabi et al., 2020[60] | Qatar | CS | 987 | 12 | RF | 0.869 | No |
| Islam et al., 2022 [61] | South Asian countries | CS | 8,18,603 | 7 | XGBoost | 0.900 | No |
| Proposed ensemble model | Ethiopia | CS | 612 | 27 | Stacking | 0.971 | Yes |

* CS: cross-sectional study

performed well in predicting HTN risk, achieving an AUC of 0.829 in an Indonesian study, indicating its potential for HTN risk assessment [56]. Furthermore, a study also employed a multi-pronged strategy for building a HTN prediction system in Malaysia, using feature selection and addressing class imbalance, achieving 74.39% accuracy with their LGBM-based model [58]. So, based on the previous discussion our proposed stacking model achieved the highest HTN prediction accuracy comparatively the existing studies (see **Table 3**). The existing study employed ML models to predict HTN risk without using clinical or genetic data in a cross-sectional study. In addition to comparing our machine learning model with other machine learning methods, it is important to consider its performance relative to classical risk prediction tools currently utilized in clinical practice, such as the Framingham Hypertension Risk Calculator [62–64]. The Framingham tools has been extensively validated in large cohorts and remains a reliable choice for long-term risk prediction, particularly for identifying individuals at risk up to four years in advance. However, it primarily relies on specific demographic and clinical variables and may lack generalizability to ethnically diverse populations, as it was developed in cohorts predominantly of European descent [62, 63]. Additionally, other HTN risk calculators, such as the Strong Heart Study, which is specifically tailored for Native-American populations, underscore the need for population-specific risk assessment tools [65]. This study focuses on risk factors prevalent in Native Americans, providing insights into hypertension prediction in a group often underrepresented in clinical studies [66]. In contrast, our ML-based model leverages advanced algorithms that integrate diverse data sources, allowing it to analyze a wider range of risk factors and capture complex interactions between them, potentially enhancing predictive accuracy. While our cross-sectional approach limits its ability to predict future hypertension onset, it identifies current risk profiles, enabling timely interventions for individuals who may otherwise be missed without a recent blood pressure check. Together, these tools highlight the importance of combining traditional clinical approaches, such as the Strong Heart Study, with advanced data-driven methods like ours, to improve hypertension prediction across diverse populations.

The SHAP analysis of the stacking model identified weight, alcohol consumption (drink), history of HTN, salt intake, age, diabetes, vegetables, BMI and fat as the key interpretable risk factors for HTN. Among these, weight and obesity (fat) were critical determinants, consistent with earlier studies showing that obese individuals are at a significantly higher risk of developing HTN compared to those with normal weight [67, 68]. Elevated BMI also emerged as a strong predictor, aligning with research linking BMI to HTN and cardiovascular diseases through mechanisms like renin-angiotensin system activation and endothelial dysfunction

[69, 70]. Dietary factors such as high salt intake and low vegetable consumption, along with alcohol consumption, were identified as significant lifestyle contributors, echoing previous finding [71, 72]. Individuals who consumed alcohol daily or after meals demonstrated a markedly higher risk of HTN compared to abstainers [73]. Additionally, age was a dominant risk factor, as older individuals (60+ years) were more likely to develop HTN compared to younger adults (18–40 years), supported by findings from Belay et al. (2022) and other systematic reviews [16, 74, 75]. This age-related risk is attributed to vascular changes, such as large artery stiffness, that occur with aging. History of hypertension (HHTN) further emerged as a significant variable, consistent with studies linking familial predisposition, shared lifestyle behaviors, and genetic factors to HTN risk [75]. Diabetes, with its bidirectional relationship to HTN, also plays a crucial role, as the two conditions share overlapping risk factors and can exacerbate each other [76]. The analysis highlights the compounded risk in obese hypertensive individuals, who have higher rates of coronary heart disease and mortality than those with either condition alone [68, 77]. Genetic predispositions, such as a higher Genetic Risk Score (GRS), were also significantly associated with increased odds of HTN, as shown in previous studies [68]. Furthermore, the relationship between salt intake and blood pressure, long established since Kempner's 1948 rice diet study, underscores the value of dietary modifications like salt restriction to mitigate HTN risk [78].

## Limitations

The dataset used in this study presents certain limitations that could affect the generalizability and integrity of the findings. With only 612 instances, the relatively small sample size may limit the robustness of the machine learning model's predictions. Additionally, there is a notable class imbalance, with 21% of the cases being HTN and 79% non-hypertensive, which could introduce bias in the results despite employing data balancing techniques like ADASYN. This imbalance may still influence model accuracy, potentially leading to overfitting. While the current study offers valuable insights, further research with larger and more balanced datasets is necessary to enhance the external validity and real-world applicability of the model's predictions. Future work could involve collaboration with larger cohorts or utilizing additional secondary data sources to mitigate these limitations.

## Conclusions

This study compared five ML algorithms and a stacking ensemble model for predicting HTN risk. The stacking model emerged as the most effective approach for identifying patients at risk of HTN. Furthermore, SHAP analysis of the stacking model revealed that physical inactivity, age, educational status, and a history of hypertension were the most significant contributing factors to HTN development. These findings suggest that the proposed integrated system can be a valuable tool in clinical settings for the early detection of patients at risk of HTN. By leveraging this information, healthcare professionals can make informed decisions that has the potential to decrease healthcare costs and improve patient outcomes. Additionally, early identification allows for the implementation of personalized interventions and targeted treatment strategies, ultimately contributing to a reduction in the overall burden of HTN in Ethiopia.

## Supporting information

**S1 Appendix. Machine learning algorithms.**
(DOCX)

**S1 Table. Demographic profiles of the respondents.**
(DOCX)

**S2 Table. Risk factors for HTN identified using five feature selection techniques.**
(DOCX)

**S3 Table. Performance evaluation of prediction models on imbalanced data.**
(DOCX)

**S1 Text. Code for machine learning-based hypertension prediction.**
(TXT)

## Acknowledgments

Authors would like to acknowledge both reviewers for their valuable comments that were helpful to improve the quality of the manuscript.

## Author Contributions

**Conceptualization:** Isteaq Kabir Sifat, Md. Kaderi Kibria.

**Data curation:** Isteaq Kabir Sifat, Md. Kaderi Kibria.

**Formal analysis:** Isteaq Kabir Sifat.

**Funding acquisition:** Md. Kaderi Kibria.

**Investigation:** Md. Kaderi Kibria.

**Methodology:** Isteaq Kabir Sifat.

**Project administration:** Md. Kaderi Kibria.

**Resources:** Md. Kaderi Kibria.

**Software:** Isteaq Kabir Sifat.

**Supervision:** Md. Kaderi Kibria.

**Validation:** Md. Kaderi Kibria.

**Visualization:** Isteaq Kabir Sifat.

**Writing – original draft:** Isteaq Kabir Sifat, Md. Kaderi Kibria.

**Writing – review & editing:** Md. Kaderi Kibria.

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
