## [Decision Letter · Decision Letter 0]

13 Sep 2024

PONE-D-24-27661Optimizing Hypertension Prediction Using Ensemble Learning ApproachesPLOS ONE

Dear Dr. Kibria,

Thank you for submitting your manuscript to PLOS ONE. After careful consideration, we feel that it has merit but does not fully meet PLOS ONE’s publication criteria as it currently stands. Therefore, we invite you to submit a revised version of the manuscript that addresses the points raised during the review process.

Please check the feedback from the reviewers. As indicated by the reviewers there are several parts of the paper that needs further clarifications.

It is particularly important to address the quality of the dataset and how well it represents the problem and task you want to solve with the approach.

It is important to address the limitations of the study and approach and compare to existing methods and related work of others. This can be tied into the discussion of the novelty of the presented approach, which needs improvement, too.

We look forward to receiving your revised manuscript.

Kind regards,

Tomo Popovic, Ph.D.

Academic Editor

PLOS ONE

**Journal requirements:**

3.  We noted in your submission details that a portion of your manuscript may have been presented or published elsewhere. [Yes, the data in this manuscript have been previously published in another paper titled 'Predicting the risk of hypertension using machine learning algorithms: A cross-sectional study in Ethiopia.' We have used this dataset and Enhance the prediction accuracy by using ensemble learning models.] Please clarify whether this publication was peer-reviewed and formally published. If this work was previously peer-reviewed and published, in the cover letter please provide the reason that this work does not constitute dual publication and should be included in the current manuscript.

4. Thank you for uploading your study's underlying data set. Unfortunately, the repository you have noted in your Data Availability statement does not qualify as an acceptable data repository according to PLOS's standards.

Reviewers' comments:

Reviewer's Responses to Questions

**Comments to the Author**

1. Is the manuscript technically sound, and do the data support the conclusions?

Reviewer #1: Yes

Reviewer #2: Partly

Reviewer #3: Partly

2. Has the statistical analysis been performed appropriately and rigorously? 

Reviewer #1: Yes

Reviewer #2: Yes

Reviewer #3: Yes

3. Have the authors made all data underlying the findings in their manuscript fully available?

Reviewer #1: Yes

Reviewer #2: Yes

Reviewer #3: Yes

4. Is the manuscript presented in an intelligible fashion and written in standard English?

Reviewer #1: Yes

Reviewer #2: No

Reviewer #3: Yes

5. Review Comments to the Author

Reviewer #1: This study focuses on predicting hypertension using the improved machine learning approach, demonstrating impressive accuracy and superiority over other known machine learning methods. This topic is crucial due to the high prevalence of hypertension, a chronic disease often unrecognized for years in some patients, leading to a high rate of complications. Early identification of risk factors and timely intervention are essential for effective prevention. The manuscript is clearly written, but several significant limitations must be addressed before it is ready for publication.

1) Study Design and Data Limitations:

The data utilized in this study were obtained through a cross-sectional design, which only reveals a temporal association between the examined factors and hypertension. Due to this study design:

• The data cannot establish a causal relationship between any examined factor and hypertension. Based solely on the presented data, the authors cannot state whether some of these factors contribute to the development of hypertension (as mentioned in line 294), if they are a consequence of it, or if their association with hypertension is purely coincidental. However, previous studies have established causal links between the described factors and hypertension. This needs to be mentioned in the discussion section as one of the important limitations of the study.

• The data cannot be used to predict whether an individual will develop hypertension in the future. Traditional methods for risk prediction, already in use in the everyday clinical practice, are able to predict the future risk over a span of 1 to 4 years. The authors need to please address this in the discussion section.

• Additionally, the authors need to discuss whether the other machine learning methods mentioned in Table 3 are based on cross-sectional data, or if they predict current or future risk.

2) Comparison with Classical Risk Prediction Tools:

The discussion section should highlight a comparison of the presented results with classical risk prediction tools currently used in clinical practice, not just other machine learning methods (e.g., the hypertension risk calculator from the Framingham study). How does this method differ from or improve upon these existing tools?

See references:

https://pubmed.ncbi.nlm.nih.gov/18195335/

https://www.ncbi.nlm.nih.gov/pmc/articles/PMC8110170/

3) Furthermore, please discuss the rationale behind calculating the current risk of hypertension. Wouldn't it be simpler and more straightforward to just measure the patient's blood pressure? Predicting future risk seems more valuable, as it could facilitate early identification of at-risk individuals and allow for timely interventions to prevent or mitigate the development of hypertension (before we can actually detect hypertension by measurement).

4) Inclusion Criteria and Population Specificity:

Please clarify the inclusion criteria used for recruiting participants in this study. Were participants randomly selected from the general population, or were there specific selection criteria? This information is crucial for readers to understand the applicability of this predictive approach to different populations. For instance, were the participants exclusively adults (older than a certain age), were female participants exclusively non-pregnant, and were elderly individuals (65 and older) included, given the increased risk of hypertension with age?

Reviewer #2: The paper titled "Optimizing Hypertension Prediction Using Ensemble Learning Approaches" addresses an important issue in healthcare: the prediction of hypertension, a leading cause of morbidity and mortality worldwide. The study employs ensemble learning methods, specifically a stacking model, to enhance predictive accuracy compared to traditional single-model approaches. The authors utilize a dataset comprising 612 participants from Ethiopia and report significant improvements in predictive performance.

The subject matter of this study is highly relevant, particularly given the global burden of hypertension and the increasing focus on personalized medicine. The authors have chosen an appropriate methodological approach by leveraging ensemble learning, which is well-suited to the complexities of hypertension prediction. Additionally, the use of SHapley Additive Explanations (SHAP) for interpreting the model's predictions is a commendable choice, as it adds a layer of interpretability, making the results more actionable in a clinical setting.

However, multiple aspects of the paper require further attention. Firstly, while the use of ensemble learning is methodologically sound, the manuscript does not introduce a significant theoretical or methodological innovation. Ensemble learning techniques, particularly stacking, are well established in the field of machine learning. For the manuscript to be considered for publication in a journal, it would benefit from demonstrating either a novel algorithm, a unique application of existing methods, or a substantial improvement over current benchmark. As it stands, the contribution of this work appears to be incremental rather than transformative. A more in-depth comparative analysis and a clearer articulation of what differentiates this approach from existing work could strengthen the paper.

Another concern is the data limitations. The study is based on a dataset of 612 participants, which is relatively small for machine learning applications. This limitation raises questions about the generalizability of the findings. Although the authors have employed data balancing techniques such as ADASYN, the small sample size could still lead to overfitting, thus limiting the external validity of the results. Expanding the dataset or providing a stronger justification for its representativeness would greatly enhance the credibility of the study.

The manuscript also faces challenges in language and clarity. There are numerous grammatical errors and instances of unusual phrasing that detract from the readability and overall professionalism of the paper. Additionally, inconsistencies in terminology, such as the incorrect spelling of "Shapley Additive Explanations," further undermine the manuscript’s presentation. A thorough revision for language clarity and grammatical accuracy is strongly recommended.

While the statistical methods employed are appropriate, they are standard. The paper would benefit from a more detailed discussion of why specific methods were chosen and how they contribute to the study’s goals. The SHAP analysis, while useful, could be explored in greater depth. The authors should consider elaborating on the clinical implications of their findings and how these results compare with established risk factors for hypertension.

While the data were sourced from a prior study, it would be good to include a brief statement regarding the ethical approval for this secondary analysis, particularly given the sensitive nature of health-related data.

This paper addresses a well-known issue in healthcare and offers a solid application of ensemble learning for hypertension prediction, but it requires significant revisions to meet the standards of a journal. Enhancing the novelty of the contribution, addressing the limitations related to data size and generalizability, and refining the language and structure of the manuscript is crucial.

Reviewer #3: The manuscript addresses an important area in healthcare analytics by using ensemble learning approaches to predict hypertension. Proposed application of multiple machine learning models, combined with feature selection and hyperparameter tuning, can further contribute to this research field.

However, there are several areas where this research falls short, particularly concerning data employed, methodology description, presentation of results, clarity and novelty of methods applied.

The manuscript lacks discussion of the limitations associated with the dataset, particularly sample size (only 612 instances), potential biases in data (21%HTN vs 79% no HTN). The quality of the dataset raises some concerns, which could impact the overall integrity of the study's findings.

The study also lacks a strong description of how the data was handled (e.g. data preparation, cleaning, outliers etc.

The reported performance metrics are notably high, especially given the small sample size. It would be beneficial to (if possible) test the stacked model on an independent dataset. The high and well balanced metrics of the proposed stacked model using ADASYN are promising, but due to the relatively small dataset, may not be representable, especially when these inconsistencies are present. Also, further discussion of how the proposed method yielded such results is needed.

The manuscript’s methodology description contains high level technical detail, particulalry in the description of the machine learning algorithms. These details can be important, but as there are no implications or discussions on potential modifications to the models, means that these descriptions may be excessive, as they primarily serve to outline the models rather than contribute to further experimentation or development. Consider simplifying or moving these discussions to the appendix.

The description of the model in focus (stacked model) should be headlighted by rigorous description and discussion. It would be beneficial to further discuss the rationale behind using Logistic Regression as the level 1 learner (does this choice assume linearity in the dataset? What happens with complex relationships?) Furthermore, why has LR been used as both level 0 and level 1 learner? What about posing other models as level 1 learners? What advantages or improvements might other models offer in terms of capturing complex relationships or enhancing the overall predictive performance?

The use of SHAP values is the strength of the manuscript, but the discussion of how these values contribute to the interpretability of the models can be expanded. Specifically, the authors should provide more insight into how the identified risk factors could be used in clinical practice to guide decision-making. From a practical viewpoint, this would enhance the relevance of the study.

Consider including an ethics statement in the manuscript, as the data used are clinical in nature.

6. PLOS authors have the option to publish the peer review history of their article (what does this mean?). If published, this will include your full peer review and any attached files.

Reviewer #1: No

Reviewer #2: No

Reviewer #3: No

---

## [Author Response · Author response to Decision Letter 0]

21 Oct 2024

Thank you for the valuable comments and suggestions provided by the reviewers and editor. We have carefully addressed each point in the revised manuscript as follows:

Reviewer #1: 

General Comment: This study focuses on predicting hypertension using the improved machine learning approach, demonstrating impressive accuracy and superiority over other known machine learning methods. This topic is crucial due to the high prevalence of hypertension, a chronic disease often unrecognized for years in some patients, leading to a high rate of complications. Early identification of risk factors and timely intervention are essential for effective prevention. The manuscript is clearly written, but several significant limitations must be addressed before it is ready for publication.

Response: Thank you so much for your important comments that help us to improve the quality of the manuscript. 

Comment 1: Study Design and Data Limitations:

The data utilized in this study were obtained through a cross-sectional design, which only reveals a temporal association between the examined factors and hypertension. Due to this study design:

• The data cannot establish a causal relationship between any examined factor and hypertension. Based solely on the presented data, the authors cannot state whether some of these factors contribute to the development of hypertension (as mentioned in line 294), if they are a consequence of it, or if their association with hypertension is purely coincidental. However, previous studies have established causal links between the described factors and hypertension. This needs to be mentioned in the discussion section as one of the important limitations of the study.

Response: Thank you for your valuable comment. We revised this statement in the revised manuscript. Please see lines 225-227, 341-344 for details. 

• The data cannot be used to predict whether an individual will develop hypertension in the future. Traditional methods for risk prediction, already in use in the everyday clinical practice, are able to predict the future risk over a span of 1 to 4 years. The authors need to please address this in the discussion section.

Response: Thank you for your comment. We have revised it in the revised manuscript accordingly. Please see the discussion section. 

• Additionally, the authors need to discuss whether the other machine learning methods mentioned in Table 3 are based on cross-sectional data, or if they predict current or future risk.

Response: Thank you for your valuable comment. We have revised it accordingly in the discussion section. Please see the Table 3 and lines 323, 324. 

Comment 2: Comparison with Classical Risk Prediction Tools:

The discussion section should highlight a comparison of the presented results with classical risk prediction tools currently used in clinical practice, not just other machine learning methods (e.g., the hypertension risk calculator from the Framingham study). How does this method differ from or improve upon these existing tools?

See references:

https://pubmed.ncbi.nlm.nih.gov/18195335/

https://www.ncbi.nlm.nih.gov/pmc/articles/PMC8110170/

Response: Thank you for your valuable suggestion. We have revised the manuscript accordingly. Please see lines 324-335 for details. 

Comment 3: Furthermore, please discuss the rationale behind calculating the current risk of hypertension. Wouldn't it be simpler and more straightforward to just measure the patient's blood pressure? Predicting future risk seems more valuable, as it could facilitate early identification of at-risk individuals and allow for timely interventions to prevent or mitigate the development of hypertension (before we can actually detect hypertension by measurement).

Response: Thank you for your valuable suggestion. We have revised it in the revised manuscript. Please see lines 281-287. 

Comment 4: Inclusion Criteria and Population Specificity:

Please clarify the inclusion criteria used for recruiting participants in this study. Were participants randomly selected from the general population, or were there specific selection criteria? This information is crucial for readers to understand the applicability of this predictive approach to different populations. For instance, were the participants exclusively adults (older than a certain age), were female participants exclusively non-pregnant, and were elderly individuals (65 and older) included, given the increased risk of hypertension with age?

Response: Thank you for your comment. We have clarified it in the revised manuscript. Please see lines 99, 100.

---

## [Decision Letter · Decision Letter 1]

7 Nov 2024

PONE-D-24-27661R1Optimizing Hypertension Prediction Using Ensemble Learning ApproachesPLOS ONE

Dear Dr. Kibria,

Thank you for submitting your manuscript to PLOS ONE. After careful consideration, we feel that it has merit but does not fully meet PLOS ONE’s publication criteria as it currently stands. Therefore, we invite you to submit a revised version of the manuscript that addresses the points raised during the review process.Please note the suggestion to further research and or consult a medical doctor to address gaps in basic knowledge on hypertension and ensure accuracy, as well as to refine certain wording and interpretations, particularly when comparing with established tools like the Framingham risk score. This is critical due to the interdisciplinary nature of the journal and your manuscript.If possible consider discussiing other risk calculators (e.g., STRONG HEART study).Please have the text checked for for grammar and style improvements.Please submit your revised manuscript by Dec 22 2024 11:59PM. If you will need more time than this to complete your revisions, please reply to this message or contact the journal office at plosone@plos.org. Please include the following items when submitting your revised manuscript:A rebuttal letter that responds to each point raised by the academic editor and reviewer(s). You should upload this letter as a separate file labeled 'Response to Reviewers'.A marked-up copy of your manuscript that highlights changes made to the original version. You should upload this as a separate file labeled 'Revised Manuscript with Track Changes'.An unmarked version of your revised paper without tracked changes. You should upload this as a separate file labeled 'Manuscript'.

We look forward to receiving your revised manuscript.

Kind regards,

Tomo Popovic, Ph.D.

Academic Editor

PLOS ONE

Reviewers' comments:

Reviewer's Responses to Questions

**Comments to the Author**

1. If the authors have adequately addressed your comments raised in a previous round of review and you feel that this manuscript is now acceptable for publication, you may indicate that here to bypass the “Comments to the Author” section, enter your conflict of interest statement in the “Confidential to Editor” section, and submit your "Accept" recommendation.

Reviewer #1: (No Response)

Reviewer #3: All comments have been addressed

2. Is the manuscript technically sound, and do the data support the conclusions?

Reviewer #1: Partly

Reviewer #3: Yes

3. Has the statistical analysis been performed appropriately and rigorously? 

Reviewer #1: Yes

Reviewer #3: Yes

4. Have the authors made all data underlying the findings in their manuscript fully available?

Reviewer #1: Yes

Reviewer #3: Yes

5. Is the manuscript presented in an intelligible fashion and written in standard English?

Reviewer #1: No

Reviewer #3: Yes

6. Review Comments to the Author

Reviewer #1: I appreciate the authors' efforts to incorporate the required changes; however, some revisions appear to be somewhat superficial and require further refinement. I recommend that they consult a medical doctor to review the text before resubmission, as the current draft contains several misinterpretations that indicate a gap in basic knowledge on the subject of hypertension. Addressing these issues would help to better showcase the hard work and dedication the authors have invested in this study.

Lines 445 to 456 (in the file with the tracked changes): I recommend a more careful choice of wording in this section. When comparing the proposed method with the established Framingham risk-prediction tool commonly used in clinical practice, it is important to note that the Framingham tool has been validated in a study of over 1,000 participants, has been in use for many years, and has demonstrated lasting reliability. Moreover, this classical hypertension risk score can predict hypertension up to four years before onset, which offers a distinct advantage over the approach in the current study. The present study, in contrast, is based on a smaller sample size and is focused primarily on predicting current hypertension status.

It’s worth acknowledging, however, that this classic tool from Frminham stuy may lack generalizability to ethnically diverse populations, as it was developed in a study focused on people of European descent. The newly proposed method could serve as a complementary approach to existing risk scores, as it identifies individuals currently suffering from hypertension who might otherwise be missed without a recent blood pressure check. The study’s results are also based on a population of different ethnicity, which may enhance its relevance across broader demographic groups.

The authors should consider discussing other hypertension risk calculators, such as the STRONG HEART study, which is particularly suited to Native American populations.

It would also be valuable to comment on whether the risk factors identified in these other studies are consistent with or differ from those proposed in this study.

Additionally, it would be beneficial for a native English speaker to review the text to improve grammar and style. For example, tin the introduction section, the term “cardiovascular disease” is missing from the second sentence, and the third sentence should use “cause” instead of “causes.”

Reviewer #3: (No Response)

7. PLOS authors have the option to publish the peer review history of their article (what does this mean?). If published, this will include your full peer review and any attached files.

Reviewer #1: No

Reviewer #3: No

---

## [Author Response · Author response to Decision Letter 1]

20 Nov 2024

Comment: Please note the suggestion to further research and or consult a medical doctor to address gaps in basic knowledge on hypertension and ensure accuracy, as well as to refine certain wording and interpretations, particularly when comparing with established tools like the Framingham risk score. This is critical due to the interdisciplinary nature of the journal and your manuscript.

Response: Thank you for your valuable comment. We consulted a medical doctor and, based on their suggestions, revised the interpretations particularly in comparisons with established tools like the Framingham risk score (see introduction section and lines 336-341).

---

## [Decision Letter · Decision Letter 2]

3 Dec 2024

Optimizing Hypertension Prediction Using Ensemble Learning Approaches

PONE-D-24-27661R2

Dear Dr. Kibria,

We’re pleased to inform you that your manuscript has been judged scientifically suitable for publication and will be formally accepted for publication once it meets all outstanding technical requirements.

Kind regards,

Tomo Popovic, Ph.D.

Academic Editor

PLOS ONE

Additional Editor Comments (optional):

Reviewers' comments:

Reviewer's Responses to Questions

**Comments to the Author**

1. If the authors have adequately addressed your comments raised in a previous round of review and you feel that this manuscript is now acceptable for publication, you may indicate that here to bypass the “Comments to the Author” section, enter your conflict of interest statement in the “Confidential to Editor” section, and submit your "Accept" recommendation.

Reviewer #1: All comments have been addressed

2. Is the manuscript technically sound, and do the data support the conclusions?

Reviewer #1: Yes

3. Has the statistical analysis been performed appropriately and rigorously? 

Reviewer #1: Yes

4. Have the authors made all data underlying the findings in their manuscript fully available?

Reviewer #1: Yes

5. Is the manuscript presented in an intelligible fashion and written in standard English?

Reviewer #1: Yes

6. Review Comments to the Author

Reviewer #1: (No Response)

7. PLOS authors have the option to publish the peer review history of their article (what does this mean?). If published, this will include your full peer review and any attached files.

Reviewer #1: No

---

## [Editor Report · Acceptance letter]

10 Dec 2024

PONE-D-24-27661R2 

PLOS ONE

Dear Dr. Kibria, 

I'm pleased to inform you that your manuscript has been deemed suitable for publication in PLOS ONE. Congratulations! Your manuscript is now being handed over to our production team.

Kind regards, 

on behalf of

Prof. Tomo Popovic 

Academic Editor

PLOS ONE